SOFTWARE

# CyVerse: Cyberinfrastructure for open science

Tyson L. Swetnam[1,¤]*, Parker B. Antin[1], Ryan Bartelme[1,2], Alexander Bucksch[1], David Camhy[3], Greg Chism[1], Illyoung Choi[1], Amanda M. Cooksey[1], Michele Cosi[1], Cindy Cowen[1], Michael Culshaw-Maurer[1,4], Robert Davey[4,5], Sean Davey[1], Upendra Devisetty[1,6], Tony Edgin[1], Andy Edmonds[1], Dmitry Fedorov[7], Jeremy Frady[1], John Fonner[8], Jeffrey K. Gillan[1], Iqbal Hossain[1], Blake Joyce[1], Konrad Lang[9], Tina Lee[1], Shelley Littin[1], Ian McEwen[1], Nirav Merchant[1], David Micklos[10], Andrew Nelson[11], Ashley Ramsey[1], Sarah Roberts[1], Paul Sarando[1], Edwin Skidmore[1], Jawon Song[8], Mary Margaret Sprinkle[1], Sriram Srinivasan[1], Dan Stanzione[8], Jonathan D. Strootman[1], Sarah Stryeck[3,9], Reetu Tuteja[1,6], Matthew Vaughn[8], Mojib Wali[3], Mariah Wall[1], Ramona Walls[1,12], Liya Wang[10], Todd Wickizer[1], Jason Williams[10], John Wregglesworth[1], Eric Lyons[1]

1 The University of Arizona, Tucson, Arizona, United States of America, 2 Pivot Bio, Berkeley, California, United States of America, 3 Graz University of Technology, Graz, Austria, 4 The Carpentries, Oakland, California, United States of America, 5 Earlham Institute, Norwich, United Kingdom, 6 Greenlight Biosciences, Durham North Carolina, United States of America, 7 ViQI Inc. Santa Barbara, California, United States of America, 8 Texas Advanced Computing Center, Austin Texas, United States of America, 9 Know Center GmbH, Graz, Austria, 10 DNA Learning Center, Cold Spring Harbor Laboratory, Long Island New York, United States of America, 11 Boyce Thompson Institute, Ithaca, New York, United States of America, 12 Critical Path Institute, Tucson, Arizona, United States of America

¤ Current address: The University of Arizona, Tucson, Arizona, United States of America
* tswetnam@arizona.edu

**Data Availability Statement:** All of CyVerse software are open source and available from their Github and GitLab organizations. https://github.com/cyverse https://gitlab.com/cyverse Data hosted in CyVerse are the property of their owners,

## Abstract

CyVerse, the largest publicly-funded open-source research cyberinfrastructure for life sciences, has played a crucial role in advancing data-driven research since the 2010s. As the technology landscape evolved with the emergence of cloud computing platforms, machine learning and artificial intelligence (AI) applications, CyVerse has enabled access by providing interfaces, Software as a Service (SaaS), and cloud-native Infrastructure as Code (IaC) to leverage new technologies. CyVerse services enable researchers to integrate institutional and private computational resources, custom software, perform analyses, and publish data in accordance with open science principles. Over the past 13 years, CyVerse has registered more than 124,000 verified accounts from 160 countries and was used for over 1,600 peer-reviewed publications. Since 2011, 45,000 students and researchers have been trained to use CyVerse. The platform has been replicated and deployed in three countries outside the US, with additional private deployments on commercial clouds for US government agencies and multinational corporations. In this manuscript, we present a strategic blueprint for creating and managing SaaS cyberinfrastructure and IaC as free and open-source software.

published datasets are given DataCite DOI and made public via the https://datacommons.cyverse.org.

**Funding:** This material is based upon work supported by the US National Science Foundation under Grant numbers: DBI-0735191 (DS was a grant recipient; TE,AE,EL,NM,PS,ES,SS,DS,JW1 received salary), DBI-1265383 (PA,NM,EL,DS,MV were grant recipients; AM,SD,UD,TE,AE,JF,BJ,TL, SL,IM,EL,NM,DM,AR,SR,PS,ES,JS,MMS,SS,DS, JDS,TS,MV,MW,RW,LW,TW,JW8,JW1 received salary), DBI-1743442 (PA,JF,EL,NM,DM,MV,TS were grant recipients; RB,IC,AC,MC,CC,MCM,SD, UD,TE,AE,JF1,JF2,JKG,IH,TL,EL,SL,IM,NM,DM,AR, SR,PS,ES,JS,MMS,SS1,DS,JDS,TS,RT,MV,RW, LW,TW,JW8,JW1 received salary). Work on CyVerse Austria was funded by the Austrian Infrastructure Program 2016/2017, Bundesministerium für Bildung, Wissenschaft und Forschung Austria, BioTechMed/Graz Hochschulraum-Strukturmittel 'Integriertes Datenmanagement.' The project was supported by Digitale TU Graz (Graz University of Technology); DC,KL,SS5,MW recieved salary. The funders did not play any role in the study design, data collection and analysis, or decision to publish or preparation of the manuscript.

**Competing interests:** The authors have declared that no competing interests exist.

This is a *PLOS Computational Biology* Software paper.

## Introduction

CyVerse, a combination of the words 'Cyber' and 'Universe', is the result of 15 years of continuous software development and 13 years in production as a free cyberinfrastructure platform for public research [1, 2]. Representing the largest and longest-running public investment in Plant and Life Science research cyberinfrastructure (117 million USD to date), it operates on free and open-source software (FOSS) and commodity hardware [3–5]. CyVerse's mission is to design, deploy, and expand a national cyberinfrastructure for life sciences research and to train scientists in its use. Here, the term 'cyberinfrastructure' encompasses software, hardware, and the people who can use and train others to operate these systems [6] (see glossary S1 Table).

Initially funded as "The iPlant Collaborative" in 2008 by the United States National Science Foundation (NSF) Directorate for Biological Infrastructure (DBI) for plant sciences and genomics research, the project expanded to support all life sciences in 2013 and rebranded as "CyVerse" in 2016. Today, CyVerse caters to various scientific domains and accelerates data-driven research by connecting public and commercial cloud and high-performance computing (HPC) platforms. It promotes open and collaborative science, reproducible scientific computing [7], and the FAIR data principles [8].

Scientific discoveries increasingly rely on data analysis using complex software systems to manage large datasets and dynamic computational workflows [9]. However, challenges arise due to the lack of specialized programming skills among domain scientists [10–12], barriers to the reproducibility of work by others [7, 13, 14], unequal access to cloud computing [15, 16], and difficulty understanding computational results [8, 17, 18]. To solve some of these problems, researchers have transitioned to cloud and HPC centers, and public research datasets have moved to commercial cloud storage [11, 12, 19–22].

The adoption of cloud-native technologies and techniques for research is currently hindered by a scarcity of educators with necessary cyberinfrastructure skills [10]. Cloud-native science involves building and storing analysis-ready data in cloud-optimized formats [12] and using cloud-native software to analyze the data on distributed computational platforms. Hybrid approaches may involve various data analysis combinations, regardless of software, operating systems, virtualization, or containers, which can run as workflows on HPC, high-throughput computing (HTC), or cloud resources [23]. Researchers must fundamentally change how they conduct and publish their data, code, and results to become cloud-native scientists.

Peer-reviewed reproducible research requires original data, software (dependencies), and instructions for running the analyses [7, 24–26]. Publishing software and analytical code are additional components beyond the requirement of publishing final datasets as part of funding agreements [27, 28]. Analysis-ready data, executable software environments (containers), and analytical code form the basis of digital 'research objects' for replicating methods or building new research on prior results and analyses [29–32]. Cyberinfrastructure, which is capable of hosting research objects, enables replication with new data and promotes reproducible science [14] (Fig 1). CyVerse focuses on two areas of open science: (i) developing spaces for creating research objects, and (ii) operating public cyberinfrastructure to reduce the effort needed to adopt cloud-native science practices.

### Scientific overview

CyVerse is one of only a few publicly available cyberinfrastructure projects which are fully open source (S2 File). Open source cyberinfrastructure levels the field, allowing contributors

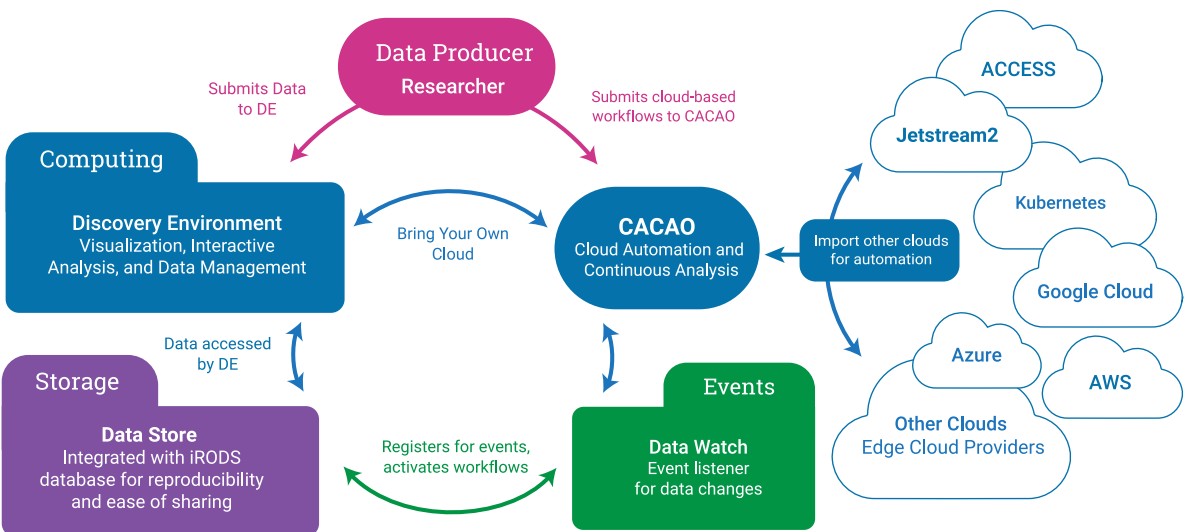

**Fig 1. Resources designed for scientists.** CyVerse offers researchers diverse compute, storage, and event-based workflow options. Data and associated metadata are uploaded and managed in an iRODS Data Store, accessible through various transfer protocols. CyVerse delivers moderate computing power for analytical research via its Discovery Environment data science workbench and connects to public and commercial cloud through the Cloud Automation and Continuous Analysis Orchestration (CACAO) platform. Researchers can register and activate workflows in CyVerse using event-based triggers through the Data Watch API.

from small or underserved institutions to contribute directly to the same science as the best equipped and well funded Research 1 scientists. CyVerse services are designed to manage research data for its entire life cycle [13, 33, 34] and to empower the FAIR [8] and CARE (collective benefit, authority to control, responsibility, and ethics) data principles [35]. By running open source software, researchers can verify every aspect of their computational workflows on the platform. By coordinating continuous analyses with development and operations ('DevOps') Fig 2, CyVerse infrastructure enables researchers to focus on their science and not on the design and deployment of bespoke infrastructure, which have higher creation and operating costs and less reusability. CyVerse education, outreach, and training (EOT) focuses on developing researchers' data science skills, the creation of "research objects", and conducting open science [36–38] via the services and products CyVerse provides.

## Technical overview

CyVerse is both Software as a Service (SaaS) and Infrastructure as Code (IaC), as well as a data hosting service [39, 40] (Fig 1). SaaS relies on the internet to connect users to centrally hosted applications. IaC uses machine-readable definition files (code) to declaratively provision computing and data storage resources rather than having to physically configure hardware and software. CyVerse uses IaC workflows based on Terraform, Ansible, and Kubernetes to enable its interactive data science workbench and cloud environments. As a SaaS, US public CyVerse is run entirely on commodity hardware on-premise (on-site), although it can also be deployed entirely on commercial cloud service providers. As IaC, CyVerse can be dynamically provisioned into larger deployments by leveraging federated Kubernetes clusters and public research cloud (e.g., Jetstream2) resources. CyVerse provides the instructions for deploying its cloud-native services via Ansible Playbooks [41], Argo Workflows [42], Kubernetes (K8s) [43], and Terraform [44] from its public code repositories (S2 Table).

**Fig 2. Layers of technology.** CyVerse software technology and cyberinfrastructure hardware components form a "layer cake" with hardware supporting services and software products. In general, the top layer is easiest to use but least flexible, while the bottom layers have the most power and utility but are least user-friendly.

CyVerse utilizes HPC resources at Texas Advanced Computing Center (TACC) and HTC through the OpenScienceGrid, each of which are connected over Internet2 [45] (Fig A in S1 Text). CyVerse is also connected to and manages commercial resources on AWS, Google Cloud, and Azure through its external partnership programs. By managing itself as SaaS and IaC, replicated versions of CyVerse can be deployed by experienced CyVerse staff in a matter of hours. Managing both SaaS and IaC has benefits from an economy of scale perspective and reduces more costly and time-consuming manual deployment processes on physical hardware. New CyVerse deployments can be undertaken by anyone using the public developer documentation at https://docs.cyverse.org.

CyVerse's customizable, multi-platform data science workbench, called the 'Discovery Environment' (DE), provides a single web-based interface for running executable, interactive, HPC and HTC applications. The DE leverages the CyVerse Terrain API, which is an externally accessible REST API with Swagger 2.0 specifications and an interactive console for most of its endpoints. The DE supports dozens of research software analysis pipelines via its HTCondor-based [46] jobs, as well as all popular integrated development environments (IDE), e.g., RStudio [47], JupyterLab [48], and Visual Studio Code [49], which are run via a Kubernetes (K8s) cluster [43]. Once container images are cached on the compute nodes, users can launch their preferred IDE in under 30 seconds.

## Design and implementation

All CyVerse APIs, SaaS, and IaC source code are provided via its public GitHub and GitLab organizations (S2 Table). DevOps and User documentation are hosted online along with manuals, tutorials, and walkthroughs (S3 Table). CyVerse's featured software stack can be best described as a 'layer cake' (Fig 2). Conceptually, each layer targets a different set of users and use cases relative to specific scientific objectives. Functionally, most researchers interact with the Products, shown in blue (Fig 2) which are supported by graphic interfaces via a web-browser. Research Software Engineers and/or DevOps personnel with advanced programming expertise can take advantage of the foundational services, shown in purple (Fig 2) and hardware resources (shown in green) for customized applications. In the S1 Text we provide additional details about CyVerse physical resources, services, and third party SaaS products which CyVerse helps manage for its user community.

## Results

CyVerse has been cited in 1,695 peer-reviewed articles as of 2023 (S1 File). CyVerse has collaborated on or enabled over 50 externally funded projects in the last five years through its "Powered by CyVerse" framework (S4 Table). Each of these leverage different aspects of CyVerse's available SaaS and IaC (S3, S4 and S5 Tables).

See S1 Text for statistics about CyVerse's tools, integrated apps (S4 Table, data store usage (Fig E in S1 Text and S6 Table), geographic user distribution (Fig J in S1 Text).

## Education outreach & training

The CyVerse Learning Center, an internal team created in 2017, has taught 48 professional workshops (both in-person and virtual) to 1,670 principal investigators, early-career faculty, post-doctoral researchers, graduate students, and professionals. Since 2014, CyVerse and community experts have presented 141 webinars on platform, software, and science topics to over 7,100 attendees, amassing more than 95,000 views on YouTube (Fig 3).

A core component of CyVerse's user-support success is rapidly answering questions from researchers, students, and learners of all career stages via a content management system (CMS). CyVerse does pay for an Intercom.io [50] account, which is integrated across CyVerse web services and into its documentation websites. Users can request support directly through Intercom's chat feature, which also generates a ticket in the Intercom system and an email to the user. While support tickets are often automatically binned for the appropriate team to respond, CyVerse staff regularly reviews tickets to ensure the correct team will answer. This internal network of communication results in CyVerse having an average response time of less than four hours to resolution for new tickets during normal working hours (Monday—Friday). Intercom also allows CyVerse staff to monitor user demographics and user experience. CyVerse staff received and answered approximately 2,000 support tickets a year over the last three years.

CyVerse was an early supporter of Software Carpentry [51, 52], now called "The Carpentries" [18, 53]. We continue to collaborate with Carpentries staff and trainers in developing novel data science training materials and in hosting workshops. Digital literacy is currently at the forefront of many universities' education and research programs, with "Data Science" institutes, colleges, departments, and degree programs being created globally to meet this demand.

Asynchronous training materials are maintained through the Learning Center website (https://learning.cyverse.org), while in-person and virtual workshops, entitled "Foundational Open Science Skills," "Container Basics," and "Advanced Containers," are offered for data

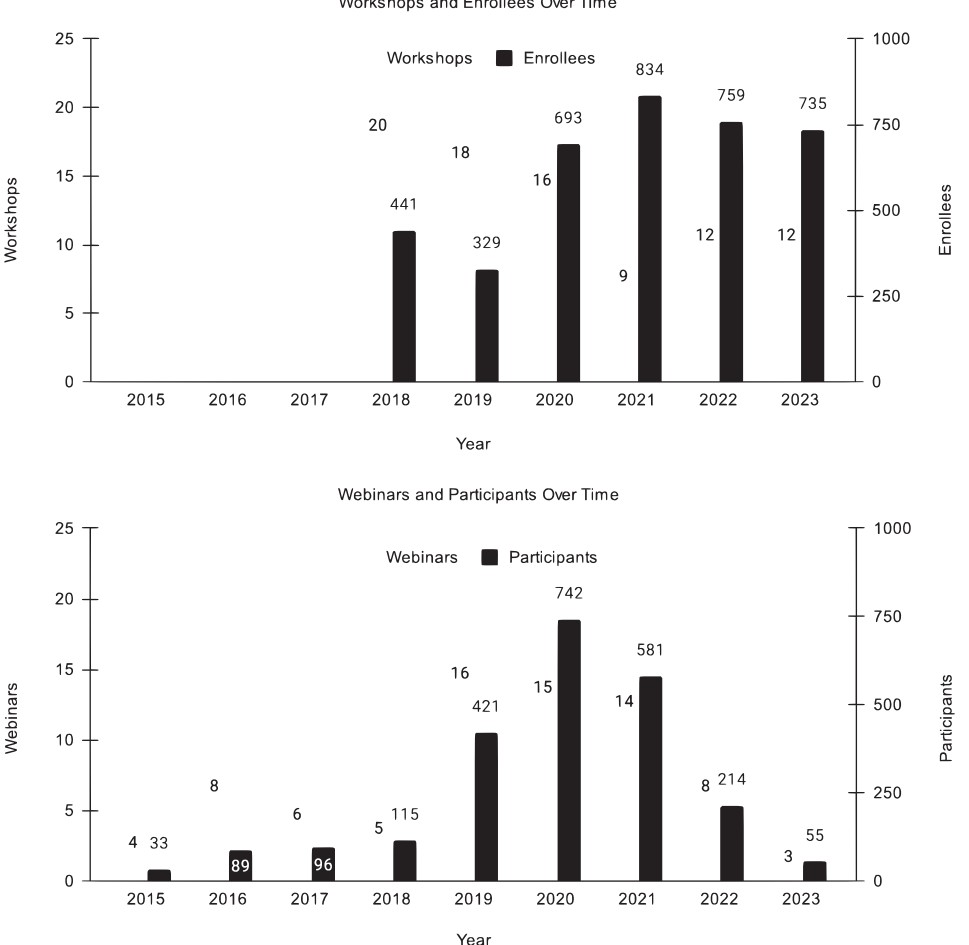

**Fig 3. Workshop and webinar participants.** In-person and Virtual Workshops and total enrollees, Virtual Webinars given and participants who RSVP'd.

scientists, educators, researchers, IT professionals, and students on a recurring basis. The workshops are designed to prepare investigators and their teams, both new and established, to meet the growing expectations of funding agencies, publishers, and research institutions for scientific reproducibility and data accessibility. Since 2018, 345 early career faculty and post-doctoral researchers have attended CyVerse Foundational Open Science Skills workshops. Numerous attendees have gone on to write successful NSF proposals, start their own research labs, or integrate open science principles into their curricula or departmental courses, using skills and knowledge gained during the workshop. A recent phenomenon includes humanities researchers enrolling in our workshops, revealing interest in these foundational computational skills from a wide range of disciplines.

## User demographics

Most CyVerse users are from a life sciences background (plant science, biology, genomics); however, over 23% of users beyond life science (Fig 4). In the last 10 years CyVerse was demonstrated to approximately 45,000 attendees at in-person or virtual workshops and webinars. Over 124,000 accounts have been created since its initial public launch in 2011. In the last four years,

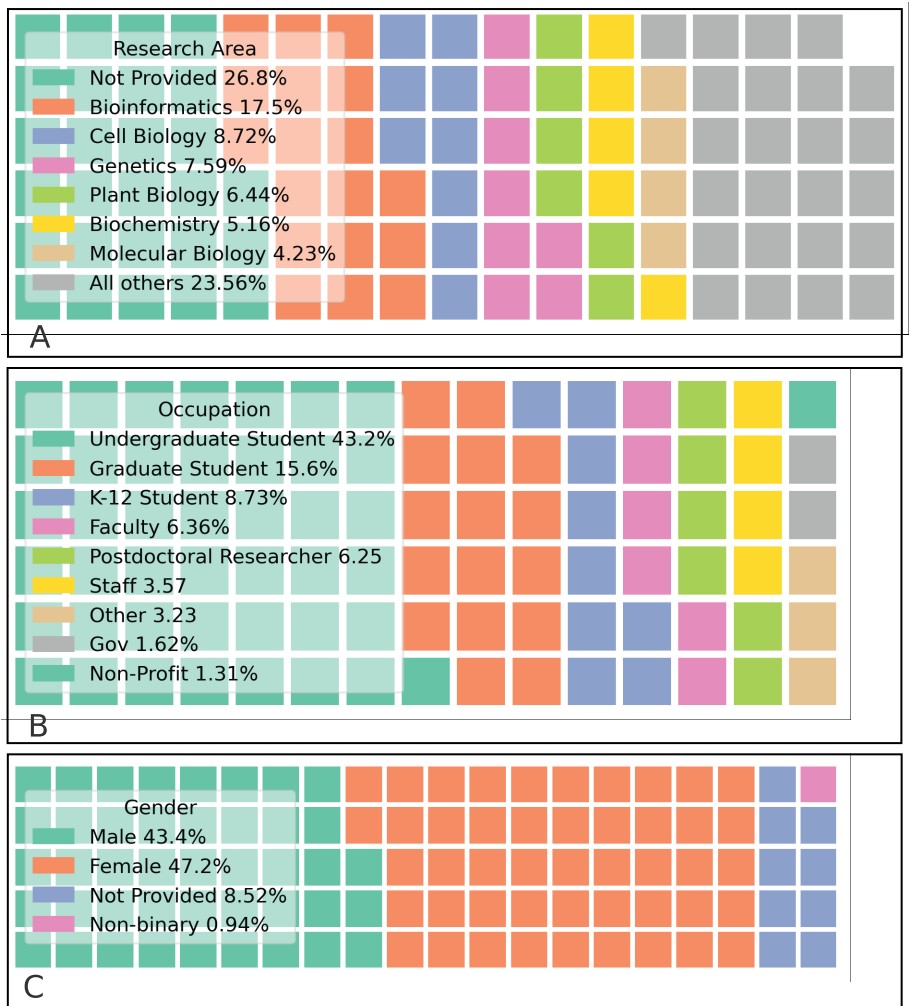

**Fig 4. Demographics registered users' research area (A), occupation (B), and genders (C).**

37,310 unique users from over 160 countries have logged into the platform (Fig J in S1 Table). The majority of users are from non-Research 1 Universities (92,275 accounts) and self-identify as being students (Fig B in S1 Text). Notably, 3.8% more users identify as female than male (Fig C in S1 Text). Additional anonymized user demographic information is provided in the S1 Text.

There are typically 40±10 unique users in web-sessions on the platform at any given time. Peak usage is observed during workshops or educational courses, with over 300 concurrent users in the Discovery Environment observed without interruption. In a typical 30-day period 1,100 ±100 registered users will log into the platform and use it at least once. Between August 2017 and August 2023 the top ten users (excluding CyVerse staff and affiliated users) launched over 1,000 web sessions; the top 100 users had 300 web sessions; the top 500 had 100 web sessions.

## Scientific impacts

CyVerse has been referenced in over 1,600 scientific publications including peer-reviewed journal articles, masters theses, and Ph.D. dissertations (S1 File). CyVerse or iPlant Collaborative are mentioned in 125 NSF awards' public abstracts ($257M funding). CyVerse staff have

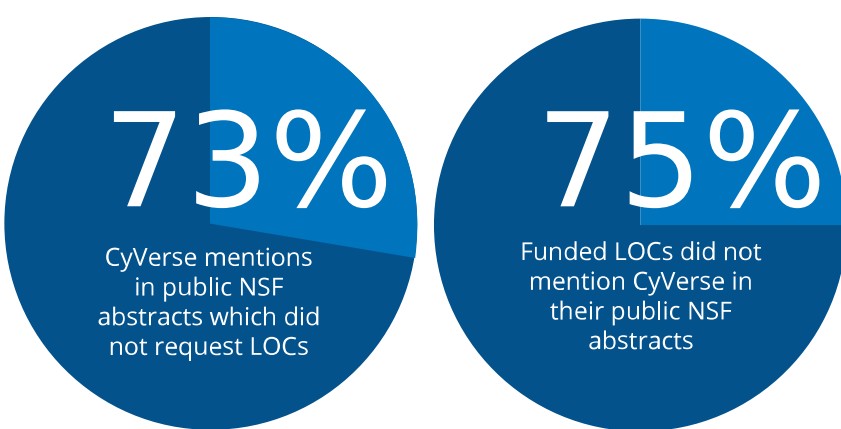

**Fig 5. Multiplier effects.** Only 25% of NSF awards which requested Letters of Collaboration (LOCs) mention "CyVerse" or "iPlant Collaborative" in their public abstract. Of the total awards that mention "CyVerse" or "iPlant Collaborative" 73% did not request LOCs.

written over 400 letters of collaboration (LOC) for grant proposals and have been a collaborating partner on over 50 federal awards to-date. We can generate a rough estimate of the total return on investment (ROI) of NSF award dollars supported by CyVerse by comparing percentage of awards for which CyVerse wrote a LOC and that mention CyVerse in their public abstract (25%), versus the total number of awards mentioning CyVerse in their public abstract without requesting a LOC (73% of all mentions) (Fig 5). Using this logic, CyVerse has likely helped to support between $700M and $1B of NSF awards. Specifically, this does not include projects supported by other US agencies (regional, state, federal, private) and international research projects. The multiplier effect on research support from NSF's $117M investment into CyVerse has likely resulted in 6× to 9× ROI.

## International expansion

In 2018, CyVerse UK was launched at the University of Nottingham [54]. CyVerse UK leverages the foundational services (described in the Design and Implementation section from CyVerse US) while maintaining their own Data Store and Discovery Environment [55]. In 2019, CyVerse Austria (CAT) was launched at Graz University of Technology (TUG) [56, 57]. CAT was initiated through the BioTechMed initiative [58] to foster collaborative, reproducible life science research in Graz. CAT connects TUG with the Medical University of Graz and the University of Graz. The system adheres to the EU-GDPR and ensures that universities comply with their respective research data management policies [59, 60], meaning that research data is stored on-site with secured access. Through the iRODS-based Data Store, researchers can work together in collaborative projects within BioTechMed. CAT has quickly grown in visibility, and recently the Austrian ministry funded a follow-up project [61] to further develop and expand CAT within Austria. In addition, given that CyVerse US, CyVerse UK, and CyVerse CAT use the same foundational technologies, our teams and user communities are able to rapidly redeploy and reuse workflows across deployments, decreasing the time it takes researchers to leverage new analytical methods. Additional international collaborations with CyVerse US are currently underway in Canada and Australia.

### Impact beyond the life sciences

Science projects beyond the life sciences leveraging CyVerse include: astronomy [62, 63], atmospheric science, national defense, earth science, environmental health [64], geology & geoscience, health science, hydrology [65], natural resources, lunar and planetary science, ocean science, oceanography, pedagogy, pollution monitoring, and space sciences [66–69] (see S1 File for the list of 1,600 publications which reported using CyVerse). As of 2023, CyVerse has provided DOIs to 153 data sets, which range from viral and plant genomes to astronomical black hole images.

CyVerse partners with numerous organizations and institutions to help develop new tools and research outside its core life science focus. Prominent examples include the DesignSafeCI for Natural Hazards, whose design was inspired by the CyVerse Discovery Environment [70], CUAHSI's HydroFrame and HydroShare which also rely on iRODS [65, 71, 72], and open workflows which leverage OpenTopography.org and the OSG [73]. CyVerse has International Traffic in Arms Regulations (ITAR) compliance for software, and has been successfully deployed on secured commercial cloud (AWS GovCloud) for defense and space situational awareness projects. CyVerse Health Information System is a HIPAA Privacy Program certified with Authority to Operate until 2026. Beyond these first degree influences that CyVerse has had, its affiliate projects, e.g., Jetstream and Jetstream2, are enabling ever more external research outcomes outside the life sciences [11, 74]. For example, the Atmosphere/Jetstream UI was adopted by the Massachusetts Open Cloud (MOC) [75] and is used for research and education.

## Availability & future directions

All SaaS and IaC templates developed by CyVerse are available on its GitHub and GitLab organizations (S2 Table) and are licensed under Open Source Initiative (OSI) compliant licenses. CyVerse developed software (e.g., iRODS CSI Driver, DE, & CACAO) are released under the BSD v2 Clause License, except in specific cases where other OSI licenses may supersede. All training material and platform documentation are licensed under the Creative Commons (CC BY 4.0) License. CyVerse operates multiple public services from the *.cyverse.org domain name service (S4 Table).

### Tomorrow's challenges

A survey of over 700 National Science Foundation (NSF) principal investigators found that a lack of skills in the use of cyberinfrastructure and in training opportunities was the greatest bottleneck to leveraging existing investments in research cyberinfrastructure by the life sciences community [10]. More broadly, this digital divide disproportionately impacts minorities and individuals working at smaller, underserved, or rural institutions, who may lack access to high speed internet and consequently, the ability to do data-intensive science [16, 76–78]. CyVerse was designed to meet this need head on, with the minimal requirement that the student or researcher has at least limited access to the internet.

The August 2022 White House Office of Science and Technology Policy (OSTP) "Nelson Memo" made clear that open science and open data are to be requirements of all federally-funded research beyond 2025 [79]. CyVerse's user registrations represent over 160 countries and dozens of scientific disciplines (Fig 3), further revealing a global demand for FOSS cyberinfrastructure. CyVerse is ready to help facilitate future open science research at any scale, having already developed services and resources explicitly around FOSS, FAIR & CARE data principles, and open access data.

After 15 years of core support from the National Science Foundation, CyVerse will be one of the largest NSF projects ever to transition to a self-sustaining revenue model. CyVerse addressed this challenge by ensuring adherence to its core vision/mission, identifying its user community, and working with established partners for advice and guidance. CyVerse has successfully launched four revenue streams. First, through an NSF supported partnership with Phoenix BioInformatics 501(c)3 in 2021 CyVerse developed and implemented a subscription system for individuals and institutions. CyVerse also continues to partner with large federally-funded research proposals in an infrastructure support role through its Powered by CyVerse program and provides on-premises deployments for institutions, organizations, and companies through its Professional Services program. CyVerse also receives funding support at the state-level from its host institutions. Together, these diversified revenue streams ensure long-term project financial stability while meeting the needs of CyVerse's diverse user communities.

As CyVerse pivots towards a sustainable model which relies in part on subscriptions for services, researchers from underserved groups are likely to be excluded at higher rates than other groups, thus widening the digital divide in research computing [76, 78]. Therefore, our goal remains to continue offering all of our services with enforced computing and storage limits for free. By taking in a diversified revenue stream we intend to maintain a 'basic' free tier for all students and for anyone interested in testing CyVerse for their work.

Treating scientific software as infrastructure, rather than as part of research, would help address the ongoing issue of sustainability of FOSS cyberinfrastructure like CyVerse, Jetstream2, and the National Research Platform. Organizations which create and support FOSS SaaS and IaC are providing services in the same vein as traditional research computing centers, which are considered necessary. If only large and wealthy institutions have the capacity and capital to invest in on-premise hardware and the people to administer it, or to pay the significant commercial cloud services fees that modern research computing requires, inequities will persist or even grow [80–83]. Therefore, it is our belief that unfettered access to research and educational cyberinfrastructure is vital to ensuring a diverse, equitable, and inclusive society [84, 85].

## Streaming toward the edge

Data intensive science from streaming data and edge computing [86–88] is one frontier at which CyVerse envisions itself in the next decade. Moving computations to the edge with the Internet of Things (IoT), Machine Learning (ML), and generative AI for remote sensing using platforms such as sUAS, and integrated sensor networks streaming real and near real-time data are all areas where CyVerse is already involved. Applying CyVerse's cyberinfrastructure capabilities to the most pressing challenges our society faces include, but are not limited to: adapting to and developing better strategies for resilience to climate change, exploring Genotype by Environment = Phenotype (G×E = P) in both agricultural and natural settings [89, 90], using ML and AI for monitoring Earth system processes and studying human health, and developing precision medicine and synthetic biological approaches to life science (See S1 Text for explicit examples).

## Supporting information

**S1 Text. Description of CyVerse core & cloud services.** Additional details about CyVerse featured platforms, core services, and cloud native services. Basemaps from Carto and OpenStreetMap CC-BY 4.0 license, (https://github.com/CartoDB/basemap-styles).
(PDF)

**S1 Table. Glossary.** Frequently used abbreviations and acronyms with descriptions.
(PDF)

**S2 Table. Version control.** Public and private version control organizations on GitHub and GitLab for CyVerse Software, Public Container Registry, and Education.
(PDF)

**S3 Table. Interfaces.** Websites under the \*.cyverse.org DNS address.
(PDF)

**S4 Table. Powered by.** External projects supported by CyVerse within the last 5 years. Resources include (Web) Hosting, Compute, Data Storage, Discovery Environment (DE), & API access.
(PDF)

**S5 Table. University of Arizona hardware.** On-premises resources maintained by CyVerse at the University of Arizona. DE = Discovery Environment, VICE = Virtual Interactive Compute Environment.
(PDF)

**S6 Table. Benchmarking Data Store transfers.** Duration of seconds within CyVerse, as well as from (download) and to (upload) other research HPC (TACC), cloud (XSEDE Jetstream2) and commercial cloud services (AWS, Google Cloud). \*Transfer duration represents a rounded number of seconds as a geometric mean of n = 30 runs.
(PDF)

**S7 Table. DE applications.** How they run, where they run, and popular applications.
(PDF)

**S1 File. Publications.** Peer-reviewed research citing the use of resources from the iPlant Collaborative (2008–2017) and CyVerse (2017-Present). Also see https://cyverse.org/publications for the latest update.
(PDF)

**S2 File. CyVerse vs others.** CyVerse's services versus other public research and commercial cyberinfrastructure. Some services offered by commercial cloud have free tiers as well as paid subscriptions. (Also see link to table).
(PDF)

## Acknowledgments

The authors are also grateful to the tens of thousands of researchers who joined the CyVerse platform and conducted their science there over the last 15 years.

## Author Contributions

**Conceptualization:** Tyson L. Swetnam, Parker B. Antin, Nirav Merchant, David Micklos, Edwin Skidmore, Sriram Srinivasan, Matthew Vaughn, Eric Lyons.

**Data curation:** Illyoung Choi, Amanda M. Cooksey, Michele Cosi, Upendra Devisetty, Tony Edgin, Blake Joyce, Tina Lee, Andrew Nelson, Reetu Tuteja, Ramona Walls.

**Formal analysis:** Tyson L. Swetnam, Upendra Devisetty, Jeffrey K. Gillan, Blake Joyce, Andrew Nelson, Jawon Song, Sarah Stryeck, Reetu Tuteja, Mariah Wall, Ramona Walls, Liya Wang, Jason Williams, Eric Lyons.

**Funding acquisition:** Tyson L. Swetnam, Parker B. Antin, John Fonner, Nirav Merchant, David Micklos, Dan Stanzione, Matthew Vaughn.

**Investigation:** Tyson L. Swetnam, Dan Stanzione, Eric Lyons.

**Project administration:** Tyson L. Swetnam, Parker B. Antin, John Fonner, Nirav Merchant, David Micklos, Mary Margaret Sprinkle, Dan Stanzione, Matthew Vaughn, Eric Lyons.

**Resources:** Alexander Bucksch, Robert Davey, Andy Edmonds, Jeremy Frady, Konrad Lang, Sarah Stryeck, Mojib Wali.

**Software:** Alexander Bucksch, Illyoung Choi, Sean Davey, Tony Edgin, Andy Edmonds, Dmitry Fedorov, Jeremy Frady, Iqbal Hossain, Ian McEwen, Ashley Ramsey, Sarah Roberts, Paul Sarando, Edwin Skidmore, Sriram Srinivasan, Jonathan D. Strootman, Mariah Wall, Todd Wickizer, John Wregglesworth.

**Supervision:** Tyson L. Swetnam, Parker B. Antin, John Fonner, Nirav Merchant, David Micklos, Andrew Nelson, Sarah Roberts, Edwin Skidmore, Mary Margaret Sprinkle, Sriram Srinivasan, Dan Stanzione, Matthew Vaughn, Ramona Walls, Eric Lyons.

**Visualization:** Iqbal Hossain, Reetu Tuteja, Mariah Wall.

**Writing – original draft:** Tyson L. Swetnam, Parker B. Antin, Ryan Bartelme, Alexander Bucksch, David Camhy, Greg Chism, Illyoung Choi, Amanda M. Cooksey, Michele Cosi, Cindy Cowen, Michael Culshaw-Maurer, Tony Edgin, Jeffrey K. Gillan, Tina Lee, Shelley Littin, Nirav Merchant, Andrew Nelson, Sarah Roberts, Paul Sarando, Edwin Skidmore, Mary Margaret Sprinkle, Sarah Stryeck, Mojib Wali, Mariah Wall, Ramona Walls, Jason Williams, Eric Lyons.

**Writing – review & editing:** Parker B. Antin.

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
