## [Decision Letter · Decision Letter 0]

7 Sep 2023

Dear Dr. Swetnam,

Thank you very much for submitting your manuscript "CyVerse: Cyberinfrastructure for Open Science" for consideration at PLOS Computational Biology. As with all papers reviewed by the journal, your manuscript was reviewed by members of the editorial board and by several independent reviewers. The reviewers appreciated the attention to an important topic. Based on the reviews, we are likely to accept this manuscript for publication, providing that you modify the manuscript according to the review recommendations.

Thank you for your submission, and my apologies that it has been difficult to secure reviews. Based on the review comments and my own reading of the text, I am content to recommend this for minor revisions, with amendments as suggested by Reviewer 1.

Sincerely,

Dan Stowell

Academic Editor

PLOS Computational Biology

Jason Papin

Editor-in-Chief

PLOS Computational Biology

Thank you for your submission, and my apologies that it has been difficult to secure reviews. Based on the review comments and my own reading of the text, I am content to recommend this for minor revisions, with amendments as suggested by Reviewer 1.

Reviewer's Responses to Questions

**Comments to the Authors:**

Reviewer #1: Comments and significant suggested edits

1) I commend the authors and their team for providing a valuable resource to the scientific community, and for their dedication to enabling its use through innovation and training.

2) The web re-design (not new I know) is an attractive improvement in form and functionality, especially the DE.

3) Supplemental - Are the metadata AVU triplets indexed for searching? If so, this would be good to include. If not, this is necessary to address specifically. Also, where do these metadata 'fit' in the description of metadata on page 20 of the supplemental material?

4) Supplemental - Page 15 - Data Commons - What is the process for curating datasets? Is there expert curation? Who performs this and how are curations quality controlled? A brief explanation would be helpful.

5) Is there a process to quality control public Apps supplied by users?

6) Fig S10 - What is 'porklock'? Is a citation needed? As written this is mysteriously uninformative.

7) Line 80 - Deployment in a matter of hours is only possible with the necessary experience and expertise. Could this statement be reworded to clarify that fast deployment does require necessary knowledge and skills and is not an 'out of the box' possibility for the majority of users? For instance, not all users can access all layers of the 'cake' stack as specified in the following section.

8) Line 147 - Are there any data (counts) available to include for the number of workshop attendees who used CyVerse as part of successful proposals and the other examples listed?

9) Scientific Impacts - an impressive section!

Minor suggested edits

1) Supplemental - Page 9 - User Portal - It may be good to specify that users must request and be approved to be powered by CyVerse. As written, it appears as though the choice is entirely in the hands of the users.

2) Line 38 - 'compliments' can work, but 'components' may be easier for a broad audience to understand

3) Line 51 - fame is less of a factor than capability - suggest changing to say something like, '...best equipped and well funded...'

4) Line 96 - Should 'CyVerse' be possessive here? 'CyVerse's featured software stack...'?

5) Line 155 - perhaps just say 'outside' or 'beyond' life sciences?

6) Be consistent with use of 'R1' and 'Research 1'

7) Supplemental - Page 20 - How challenges are met - Should this section be moved to the main manuscript? Especially the first paragraph. The last 2 paragraphs appear a little awkwardly fitted in.

**Have the authors made all data and (if applicable) computational code underlying the findings in their manuscript fully available?**

Reviewer #1: Yes

PLOS authors have the option to publish the peer review history of their article (what does this mean?). If published, this will include your full peer review and any attached files.

Reviewer #1: No

Figure Files:

Data Requirements:

Reproducibility:

References:

---

## [Decision Letter · Decision Letter 1]

27 Nov 2023

Dear Dr. Swetnam,

We are pleased to inform you that your manuscript 'CyVerse: Cyberinfrastructure for Open Science' has been provisionally accepted for publication in PLOS Computational Biology.

Best regards,

Jason Papin

Editor-in-Chief

PLOS Computational Biology

Reviewer's Responses to Questions

**Comments to the Authors:**

Reviewer #1: This is an excellent manuscript describing a valuable resource. I have no other recommendations.

**Have the authors made all data and (if applicable) computational code underlying the findings in their manuscript fully available?**

Reviewer #1: Yes

PLOS authors have the option to publish the peer review history of their article (what does this mean?). If published, this will include your full peer review and any attached files.

Reviewer #1: No

---

## [Editor Report · Acceptance letter]

1 Feb 2024

PCOMPBIOL-D-23-00939R1 

CyVerse: Cyberinfrastructure for Open Science

Dear Dr Swetnam,

I am pleased to inform you that your manuscript has been formally accepted for publication in PLOS Computational Biology. Your manuscript is now with our production department and you will be notified of the publication date in due course.

With kind regards,

Judit Kozma
